# Multiple Roles of Black Raspberry Anthocyanins Protecting against Alcoholic Liver Disease

**DOI:** 10.3390/molecules26082313

**Published:** 2021-04-16

**Authors:** Ting Xiao, Zhonghua Luo, Zhenghong Guo, Xude Wang, Meng Ding, Wei Wang, Xiangchun Shen, Yuqing Zhao

**Affiliations:** 1The State Key Laboratory of Functions and Applications of Medicinal Plants, The Department of Pharmaceutic Preparation of Chinse Medicine, The High Educational Key Laboratory of Guizhou Province for Natural Medicianl Pharmacology and Druggability, School of Pharmaceutical Sciences, Guizhou Medical University, Guiyang 550025, China; tingjinxiao@126.com; 2School of Functional Food and Wine, Shenyang Pharmaceutical University, Shenyang 110016, China; zhonghua_2398@163.com (Z.L.); xudewanglnu@gmail.com (X.W.); Meng_Ding2019@163.com (M.D.); wangweiash@163.com (W.W.); 3The Key Laboratory of Optimal Utilizaiton of Natural Medicine Resources, School of Pharmaceutical Sciences, Guizhou Medical University, University Town, Guian New District, Guiyang 550025, China; 4School of Pharmacy, Guizhou University of Traditional Chinese Medicine, Guiyang 550025, China; guo_zhenghong@163.com; 5Key Laboratory of Structure-Based Drug Design and Discovery of Ministry of Education, Shenyang Pharmaceutical University, Shenyang 110016, China

**Keywords:** black raspberry, anthocyanins, alcoholic liver injury, antioxidant, apoptosis

## Abstract

This study aimed to investigate the protective effect of black raspberry anthocyanins (BRAs) against acute and subacute alcoholic liver disease (ALD). Network analysis and docking study were carried out to understand the potential mechanism. Thereafter, the serum biochemical parameters and liver indexes were measured, the histopathological changes of the liver were analyzed in vivo. The results showed that all tested parameters were ameliorated after the administration of BRAs with alcohol. Meanwhile, there was increased protein expression of NF-κB and TGF-β in extracted livers, which was associated with hepatitis and hepatic fibrosis. Furthermore, BRAs and cyanidin-*3*-*O*-rutinoside exhibited cytotoxic effects on t-HSC/Cl-6, HepG2, and Hep3B and induced the apoptosis of HepG2 cells; downregulated the protein expression level of Bcl-2; upregulated the level of Bax; and promoted the release of cytochrome C, cleaved caspase-9, cleaved caspase-3, and cleaved PARP in HepG2 cells. In addition, the antioxidant activity of BRAs was tested, and the chemical components were analyzed by FT-ICR MS. The results proved that BRAs exert preventive effect on ALD through the antioxidant and apoptosis pathways.

## 1. Introduction

Long-term alcohol consumption damages the digestive system, circulatory system, urinary system, and blood system [1]. The liver is the first organ responsible for detoxification. Thus, alcohol abuse can cause alcoholic liver disorders (ALD), such as fatty liver, alcoholic hepatitis, liver fibrosis, and cirrhosis, and increases the incidence of hepatocellular carcinoma [2,3]. According to a 2018 WHO report, harmful use of alcohol kills more than 3 million people each year and causes more than 5% of the global disease burden [4]. Excessive alcohol consumption has been the main cause of liver morbidity and mortality worldwide [5], along with globalization-associated economic and lifestyle changes. The most important and effective treatment for ALD is abstinence; according to the stage of the disease, the treatment plan varies, such as nutritional therapy, pharmacological therapy, psychotherapy, and liver transplant. The chemical drugs used to alleviate ALD have shown side effects. Thus, it is important to find alternative active substances from natural sources, such as plants and fruits.

Anthocyanins are the largest group of water-soluble polyphenolic pigments in the plant kingdom and responsible for the bright colors (red, purple, or blue) of flowers, skin, seeds, fruits, and leaves [6]. They have strong antioxidant, antiaging, anti-inflammatory, antidiabetic, and anticancer properties and may be used as natural alternatives to synthetic food colors [1,7]. In recent years, anthocyanins have been reported to protect the liver from various kinds of damage [8]. For example, anthocyanins from purple sweet potato and black rice have the preventive effect on acute or subacute ALD [1,9]. Also, anthocyanins can ameliorate or inhibit non-alcoholic steatohepatitis or liver damage induced by cyclophosphamide, acrylamide, endotoxin, d-galactosamine, palmitic acid, and CCl_4_ [10,11,12,13,14].

Black raspberry (*Rubus occidentalis*) belongs to the family Rosaceae. It is rich in various bioactive phytochemical constituents, especially anthocyanins, which have potential health-promoting effects, including antioxidant, anticancer, anti-inflammatory, and antiangiogenic activities as natural products [15]. Therefore, black raspberry exhibits strong antioxidant, anticancer, anti-inflammatory, and antiangiogenic effects [16,17]. In addition to fresh natural products, black raspberry can also be processed into a variety of foods, such as fruit juice, jelly, refreshing drink, sugar-stained fruit, jam, fruit wine, and fruit syrup. A recent report indicated that ethanol/H_2_O extracts of black raspberry can inhibit Concanavalin A-induced liver injury and the hepatic protection is associated with a decrease in lipid peroxidation and NDA oxidative damage [18]. However, there is no research on the protective effect of anthocyanins from black raspberry on liver disease induced by long-term alcohol consumption.

Network pharmacology is a novel method based on the development of “active component-target-disease” interaction network multidisciplinary fusion theory. Molecular docking is available for modeling interactions between small molecules and proteins. In this study, a network pharmacology study of black raspberry (BRAs) was established through molecular docking and network analysis. Furthermore, anthocyanin-rich fractions from BRAs were detected by the HPLC-FT-ICR MS method. Alcoholic liver injury in vivo experiments were carried out to investigate the potential protection of BRAs for the first time by establishing subacute and acute alcoholic liver disease models. Cytotoxic activities of t-HSC/Cl-6, HepG2, and Hep3B cells and HepG2 cell apoptosis induced by BRAs and cyanidin-*3*-*O*-rutinoside were measured.

## 2. Results and Discussion

### 2.1. GO Functional Enrichment and Pathway Analysis

Network pharmacology, a key technology of system biology, has attracted much attention for researching the molecular mechanism of complicated diseases [19]. As shown in Figure 1a–l, the targets of PPI network at top 5 were mainly involved in the metabolic process of response to oxidative stress, cellular response to chemical stress, cellular response to oxidative stress, protein kinase B signaling, regulation protein kinase B signaling, multicellular organismal homeostasis, organic anion transport, and protein autophosphorylation. Cellular component (CC) analysis showed that the targets are mainly related to membrane raft, membrane microdomain, membrane region, and cell projection membrane at top 3. Based on the molecular function (MF) analysis, it could be seen that the targets at top 4 mainly involve protein serine/threonine kinase activity, protein tyrosine kinase activity, transmembrane receptor protein tyrosine kinase activity, transmembrane receptor protein kinase activity, lyase activity, and carbon-oxygen lyase activity. As shown in Figure 1d,h,l, the Kyoto Encyclopedia of Genes and Genomes (KEGG) pathway enrichment analysis suggested that the targets mainly related to signal pathways, such as the PI3K-Akt signal pathway, proteoglycans in cancer, Rap1 signal pathway, EGFR tyrosine kinase inhibitor resistance, nitrogen metabolism, and focal adhesion at top 4. The results indicated that the action targets of cyanidin-*3*-*O*-rutinoside were distributed in different metabolic pathways. The “multitarget and multipathway” mutual regulation is the possible mechanism for the treatment of ALD.

### 2.2. Docking Results of Anthocyanins

As an important tool in computer-aided molecular design, molecular docking was used to predict the biological experiment by simulating the interaction of small molecules and biological macromolecules. The binding energy and binding modes of the three anthocyanins with Bcl-2, Caspase 9, and Cytochrome c are shown in Table 1 and Figure 2. The lower binding energy indicates easier binding with the protein. Therefore, for Bcl-2, Caspase 9, and Cytochrome c, cyanidin-*3*-*O*-xylosylrutinoside might be more likely to bind and was observed to be well fitted into the binding domain of Bcl-2, Caspase 9, and Cytochrome c. For Bcl-2, Caspase 9 and Cytochrome c, cyanidin-*3*-*O*-xylosylrutinoside had the binding energy of −8.3, −8.3, and −7.7 kcal/mol, respectively. Conversely, as the number of sugars went up, anthocyanin had the lowest binding energy.

### 2.3. Antioxidant Activities

It is known that oxidative stress is involved in more than 100 disease conditions in animals and humans, including alcohol-induced hepatotoxicity [20]. The reactive oxygen species (ROS) are generated during the metabolism of alcohol to acetaldehyde by alcohol dehydrogenase, acetaldehyde dehydrogenase, and cytochrome P450 in liver. The excessive generation of ROS results in an imbalance between the oxidant and antioxidant systems, lipid peroxidation, inflammation, liver cell damage, and even apoptosis [21]. Therefore, the efficacy of protective constituent against ALD essentially depends on its capacity of either reducing the harmful effects (ROS) or maintaining the normal physiology of cells and tissues [9]. Anthocyanins have been shown to exert oxidative-stress-associated functional protein modulation through various signaling pathways [22]. Hence, anthocyanins are considered to have the potential of protective action in liver injury.

Unlike other studies, the BRAs were purified by chromatography, and the content of anthocyanins was higher than ethanol/H_2_O extracts. The antioxidant activity of BRAs was evaluated by 2,2-diphenyl-1-picrylhydrazyl (DPPH), ferric reducing antioxidant power (FRAP), and 2,2′-azino-bis (3-ethyl-benzothiazoline-6-sulfonic acid) (ABTS). In the DPPH assay, the BRAs showed radical scavenging activity with EC_50_ was 3.95 ± 0.08 μg/mL and the positive control (vitamin C) was 5.13 ± 0.12 μg/mL. The antioxidant activity was higher than that of crude polyphenols with EC_50_ 8.13 ± 0.11 μg/mL. BRAs had good activity with 6.13 ± 0.10 μg/mL in ABTS and 2.11 times higher than the positive control in the FRAP assay. Crude polyphenols had the EC_50_ of 35.74 ± 3.37 μg/mL in ABTS and only 0.57 times compared with the positive control (12.30 ± 0.21 μg/mL in the ABTS assay). The result suggests that anthocyanins from black raspberry show good antioxidant activity.

### 2.4. FT-ICR MS Analysis

Polyamide column chromatography is an effective method for the separation and purification of flavonoids (anthocyanins). In this experiment, the total monomeric anthocyanin contents of crude extract were 7.32 ± 2.19 mg/g, and the BRAs were 41.24 ± 4.72 mg/g by the pH differential method.

FT-ICR MS is an ideal method for profiling anthocyanins with high front-end resolution and analyze isobaric species with mass less than that of an electron [23]. The key features of the FT-ICR MS analysis include high mass accuracy measurements and the ability to resolve isobaric species with mass less than that of an electron [24]. Seven known anthocyanins were identified in black raspberry by HPLC-FT-ICR MS/MS, which was consistent with a previous article [15], and cyanidin derivatives were the main anthocyanins. The information about the peaks and the identification of anthocyanins are demonstrated in Table 1 and Figure 1. In the positive ion mode, peaks 1–4 produced a fragment ion with *m*/*z* of 287.05, which indicated aglycone ([cyanidin]^+^). Peaks 5 and 6 had [M^+^] ion with *m*/*z* of 433.11031 (0.16 ppm) and 579.16687 (1.46 ppm); the MS/MS product ion scan produced a fragment ion with *m*/*z* of 271.05, showing that the anthocyanin contained aglycone ([pelargonidin]^+^) within its structure. Peak 7 with *m*/*z* 609.18196 of molecular and fragment peonidin aglycone ion with *m*/*z* 301.07027 can be identified as peonidin-*3*-*O*-rutinoside (Figure 3 and Table 2).

Two main anthocyanins in BRAs were cyanidin-*3*-*O*-rutinoside and cyanidin-3-*O*-glucoside, and the contents were measured by HPLC with 23.71 ± 6.33 mg/g and 3.92 ± 0.71 mg/g, respectively.

### 2.5. Measurement of Biochemical Parameters

Liver injury further causes hepatomegaly, bradygenesis, and liver cirrhosis, while the liver index can reflect the damage to the liver [28]. The liver weights and liver index of the model group increased perhaps due to the retention of macromolecules by alcohol in the cells that constitutively secrete protein and eventually lead to hepatomegaly or “balloon,” a hallmark of alcoholic liver disease [29]. The effects of BRAs on growth performance and liver indexes are shown in Table 3. Compared with the model group, the liver weight and the liver index of BRA-treated groups reduced significantly. Especially, in the high-dose BRA group, the liver weight and the liver index reduced by 13.64% and 14.59%, respectively. In the acute ALD experiment, the liver weight and the liver index reduced by 17.48% and 18.44%, respectively, which was better than the subacute ALD experiment. However, in the subacute and acute ALD experiments, there was a little difference for the liver weight among the low-, middle-, and high-dose BRAs. These results were inconsistent with the report by Cai et al. [30].

Alcohol abuse damages cell membrane permeability and the cellular enzymes stored in the liver leak into plasma [31]. As important diagnostic markers for liver function, the ALT, AST, CHO, LDL, and TBIL levels have a close relationship with liver cell membrane damage, metabolic syndrome, and lipid metabolism in the liver [1,32]. In acute or subacute ALD experiments, the levels of these markers were increased in the model groups. In the protective experiments of BRAs, BRA-treated groups (low-, middle-, and high-dose groups) all reversed these functional markers toward near normal, and reduction was dose dependent. The protective effect of BRAs is shown in Table 3. These results indicated that the co-administration of BRAs significantly suppressed liver damage, resulting in less release of these markers from the liver tissues into the blood. The possible reason may be that anthocyanins exhibit strong antioxidant activity, and their inhibition of oxidative stress contributes toward protection from alcohol-induced liver injury [33]. This is the first attempt to evaluate the role of black raspberry in attenuating alcoholic liver injury in vivo. The results were in accordance with the conclusion reported by Sun and coworkers [1].

### 2.6. Histopathological Observation

Histopathological observation provided the direct evidence of the protective effects. Ballooning degeneration, hydropic degeneration, inflammation, hepatic sinus dilated edema, and spotty necrosis occur in alcoholic-induced liver disease. The histopathology of subacute ALD mice is shown in Figure 4A. The liver section from the control group (Figure 4Aa) showed complete and clear cytoplasm, portal vein, and prominent nucleolus. The liver section from the model group showed extensive ballooning degeneration (Figure 4Ab). It was found that the administration of BRAs reversed this liver damage. There was less ballooning degeneration in the low-dose BRA group (Figure 4Ac), and liver damage also improved except slight inflammation in the middle-dose BRA group (Figure 4Ad). In the high-dose BRA group (Figure 4Ae), the tissue conditions were similar to the observation in the control group, which showed a greater effect than in the low- and middle-dose BRA groups. The histopathology of acute ALD mice is shown in Figure 4B. Compared with the control group, the liver section from the model group showed extensive ballooning degeneration and hepatic sinus dilated edema (Figure 4Bb), which alleviated significantly in low- and middle-dose BRA groups and no pathological changes were observed in the high-dose BRA group.

### 2.7. Western Blotting Analysis

The oxidative damage induced by alcohol can activate liver inflammatory response, leading to the release of a large number of cytokines and inflammatory mediators [34]. NF-κB can be directly activated by oxidative stress and lead to the excessive synthesis of TNF-*α* and promote hepatocyte apoptosis [35,36]. At the same time, Kupffer cells, sinus endothelial cells, and necrotic liver cells produce a large amount of TGF-β through autocrine or paracrine, leading to the accumulation of large amounts of extracellular matrix, resulting in liver cell death and tissue fibrosis [37]. In this study, we examined the expression levels of NF-κB and TGF-β in ALD mice (Figure 4C). The expression levels of NF-κB and TGF-β were downregulated after treatment with BRAs in the acute and subacute experiments, whereas in the model group, there was a significant increase in the expression of both proteins. These results suggested that anthocyanins from BRAs can be good regulators against inflammation and fibrosis injury in ALD mice.

### 2.8. Anthocyanins in Black Raspberry-Induced Apoptosis on HepG2 Cells

#### 2.8.1. The Antiproliferative Effect of Anthocyanins in Black Raspberry

Epidemiological evidence suggests that liver fibrosis and cancer are closely related to alcohol abuse [3]. Liver fibrosis is a wound-healing response elicited by numerous liver-damage agents and occurs in the early stages of alcoholic liver injury, and late stage often turns into liver cancer [38]. Liver cancer is an aggressive malignant disease with a poor prognosis [39]. The cytotoxicity of BRAs and cyanidin-*3*-*O*-rutinoside was determined in the GSE-1 culture model for 48 h of incubation. In the concentrations tested, no significant toxicity was noted. The cytotoxic activity of BRAs and cyanidin-*3*-*O*-rutinoside for t-HSC/Cl-6, HepG2, and Hep3B cells was evaluated using the 3-(4,5-dimethylthiazol-2-yl)-2,5-diphenyltetrazolium bromide (MTT) assay (Table 4). BRAs showed cytotoxic activities to t-HSC/Cl-6, HepG2, and Hep3B with IC_50_ values of 202.91 ± 10.17, 198.63 ± 9.68, and 181.00 ± 12.34 μg/mL, respectively. Intriguingly, cyanidin-3-*O*-rutinoside showed perfect cytotoxic activities to t-HSC/Cl-6, HepG2, and Hep3B with IC_50_ values of 30.04 ± 3.54, 22.62 ± 2.89, and 16.73 ± 2.37 μg/mL, which were more effective than positive-control Silymarin and Mitomycin C. These results suggested that the anthocyanins from black raspberry may have protective activity against liver fibrosis and cancer.

#### 2.8.2. Morphological Changes in HepG2 Cells

Apoptosis plays a crucial role in the eradication of cancer cells [40]. Changes in cells morphology may give an intuitive concept of apoptosis. To examine whether BRAs and cyanidin-*3*-*O*-rutinoside induced apoptosis in HepG2 cells, we used DAPI/PI staining to observe morphological changes of the cell nucleus. Compared with the control group, both BRAs and cyanidin-*3*-*O*-rutinoside showed significant morphological changes in response to treatment when followed for 18 h by a fluorescence microscope (Figure 4D). The proportion of viable HepG2 cells was decreased in a time- and dose-dependent manner, and cells were condensed chromatin nucleus cultured with BRAs and cyanidin-*3*-*O*-rutinoside, including the appearance of membrane blebbing and granular apoptotic bodies, which are characteristics of cell apoptosis.

#### 2.8.3. Anthocyanins Induce Apoptosis in HepG2 Cells

The imbalance between cell proliferation and death significantly influences the development of the reproductive system, such as during folliculogenesis and spermatogenesis [41]. Apoptosis plays an important role in germ cell stages in tumors, and the inhibition of tumor growth always self-limits the cancer proliferation through the mechanism of apoptosis [42]. Annexin V-FITC assay revealed BRAs and cyanidin-*3*-*O*-rutinoside also induced apoptosis in a dose-dependent manner in HepG2 cells. Q2 and Q4 stand for late and early apoptotic cells, respectively (Figure 4E). The results indicated that BRAs (120, 240, 480 μg/mL) induced cells apoptosis of 8.2%, 16.7%, and 49.6%, respectively. In the same way, cyanidin-*3*-*O*-rutinoside (25, 50, 100 μM) induced cells apoptosis of 7.1%, 9.6%, and 20.5%, respectively.

#### 2.8.4. Effects of Anthocyanins on Expression of Cell Apoptosis-Related Proteins

To further investigate the potential molecular mechanism of apoptosis induced by BRAs and cyanidin-*3*-*O*-rutinoside, western blotting was performed to detect apoptotic protein expression. The antiapoptosis proteins (Bcl-2, Bcl-XL) and the pro-apoptosis proteins (Bax, Bad, Bak) belong to the Bcl-2 family, which can balance the change of △*Ψ*m to regulate the intrinsic apoptotic pathway via mitochondrial apoptotic pathway [43]. Furthermore, the activated caspase family are the central part of apoptosis in the programmed cell death signal network. The functional apoptosome was assembled by the interaction of Apaf-1, pro-caspase-9, and cytochrome c in the cell cytosol, and the downstream executioner caspase-3 accompanied with the cleavage of PARP was subsequently activated [44].

In our study, the expression level of antiapoptosis proteins Bcl-2 was downregulated and the expression level of pro-apoptosis protein Bax was upregulated after the cells were treated with BRAs and cyanidin-*3*-*O*-rutinoside. These results indicated that anthocyanins from black raspberry may promote the apoptosis of cancer cells via mitochondrial apoptotic pathway. Subsequently, the release of cytochrome c, cleaved caspase-9, cleaved caspase-3, and cleaved PARP was upregulated as apoptotic cells increased (Figure 4F). All the results suggested that BRAs and cyanidin-*3*-*O*-rutinoside manifest good efficacy in inhibiting cancer cells proliferation by inducing cell apoptosis.

## 3. Materials and Methods

### 3.1. Plant Materials and Chemicals

Black raspberry was purchased from Blackberry Seedlings Cooperatives, Jilin, China. The fresh berries were washed, freeze dried under vacuum, and crushed into powder. 2,2-diphenyl-1-picrylhydrazyl (DPPH), 2,2′-azino-bis (3-ethyl-benzothiazoline-6-sulfonic acid) (ABTS), 2,4,6-tris(2-pyridyl)-s-triazine (TPTZ), and Folin–Ciocalteu reagent were purchased from Sigma-Aldrich (St. Louis, MO, USA). Ascorbic acid (V_C_) and polyamide resin were purchased from Sinopharm Chemical Reagent Co., Ltd. Cyanidin-*3*-*O*-glucoside and cyanidin-*3*-*O*-rutinoside (>90% by HPLC) were isolated in our previous work. Fetal bovine serum, Dulbecco’s Modified Eagle’s medium (DMEM), and penicillin-streptomycin were purchased from Thermo Fisher Scientific Co., Ltd. (St. Wyman, MA, USA). MTT (3-(4,5-Dimethylthiazol-2-yl)-2,5-diphenyltetrazolium bromide) reagent, Mitomycin C and Silymarin were acquired from Sigma-Aldrich (St. Louis, MO, USA). Acetonitrile, methanol and formic acid of HPLC grade were purchased from Fisher Scientific (Pittsburgh, PA, USA).

### 3.2. Gene Ontology and KEGG Analysis

The Gene Ontology (GO) database (http://geneontology.org/ (accessed on 16 August 2020)), which includes molecular function (MF), biological process (BP), and cellular component (CC), could be used to identify the biological mechanisms of high-throughput genomic or transcriptome data. The compound was cyanidin-*3*-*O*-rutinoside, and the targets were hepatitis, hepatic fibrosis and liver cancer. The Kyoto Encyclopedia of Genes and Genomes (KEGG) database (https://www.kegg.jp/ (accessed on 26 July 2020)) could identify the function and biological correlation of the candidate targets [45,46]. In this study, the “ClusterProfiler” package in R was used for the GO and KEGG pathway analysis. Among these enrichment analysis results, the Y-axis represents the name of the pathway, and the X-axis represents the enrichment factor (Generation, the number of genes belonging to the pathway in the target gene/the number of all the pathway genes in the background gene set). The bubble size represents the number of genes belonging to the pathway in the target gene. The bubble color represents the enrichment significance, that is, the value of p.adjust. The threshold was set as *p*-value < 0.01.

### 3.3. In Silico Molecular Docking Study

The in silico molecular docking was based on the top 1 of the drug-target network KEGG pathway rank. The PI3K/Akt signaling pathway plays an important role in biological processes such as protein synthesis, metabolism, cell cycle regulation, proliferation, and apoptosis [47]. The PI3K/Akt signaling pathway regulates the proliferation and survival of tumor cells and plays a crucial role in the process of apoptosis [48]. Studies have shown that many active components of medicinal plants can induce apoptosis in cancer cells by inhibiting the PI3K/Akt signaling pathway [49].

To determine the potency of three main anthocyanins (cyanidin-*3*-*O*-glucoside, cyanidin-*3*-*O*-rutinoside, and cyanidin-*3*-*O*-xylosylrutinoside) in black raspberry as an inhibitor against Bcl-2 and Caspase 9, an activator of Cytochrome c, a molecular docking in silico study was carried out to predict the structure of a ligand within the constraints of a receptor-binding site [50]. The crystal structures of Bcl-2, Caspase 9, and Cytochrome c were downloaded from PDB with IDs 4LVT, 4RHW, 2BC5. ChemBioDraw (PerkinElmer, Cambridge, MA, USA) and AutoDock Tools (ADT, version: 1.5.6) are the docking programs used, and Discovery Studio 2016 Client is used for structure exhibition. Blind docking over the whole was carried out using Genetic Algorithm, and the resultant complex structures were investigated by using the conformations of the most favorable binding energy.

### 3.4. Preparation of Black Raspberry Anthocyanins

Freeze-dried powder (20 g) was extracted through ultrasonication by using ethanol–water (75:25, *v*/*v*) at 35 °C for 3 h. The acquired crude extract was filtered through a Buchner funnel. A rotary evaporation apparatus was used to remove the solvent under vacuum. The extracted crude polyphenols were redissolved in 60 mL of 10% ethanol containing 0.1% HCl and loaded onto a polyamide chromatography column (5 mm × 72 mm, 42 g). The column was washed with 10% ethanol to remove sugars. Subsequently, the absorbed anthocyanins were eluted with 30% ethanol. The solvent was recovered to obtain BRAs, which were stored at −20 °C prior to use. All processes were performed in the dark. Anthocyanin contents were verified by the pH differential method in accordance with previous reports [12]. The results were expressed as mg cyanidin-*3*-O-glucoside equivalents/g dry weight (DW).

### 3.5. Antioxidant Activities

The antioxidant activities of crude polyphenols and BRAs were determined on the basis of free radical scavenging capacity (DPPH and ABTS) and ferric reducing antioxidant power (FRAP). DPPH, ABTS, and FRAP were evaluated using the methods previously described by Sá [51], Re [52], and Sun [53], with slight modifications.

### 3.6. HPLC-FT-ICR MS/MS Analysis

The anthocyanins in BRAs were analyzed with an Agilent 1260 HPLC system coupled with a quaternary pump, an online degasser, an autosampler, a thermostatically controlled column compartment, and a diode-array detector. The system was connected to a FT-ICR MS and a 7.0 T superconducting magnet (Bruker Daltonics, Bremen, Germany). An Agilent HC-C18 column was applied at a flow rate of 0.8 mL/min. The mobile phase was 1% formic acid in acetonitrile (A) and 1% formic acid in water (B). The process utilized a linear gradient with 10% A in B for 7 min followed by a linear gradient to 28% A in B at 40 min with a column regeneration time of 10 min between injections and an injection volume of 15 µL. The column temperature was maintained at 25 °C, and the detection wavelength was 280 nm. UV spectra were also recorded from 190 nm to 400 nm for peak characterization.

### 3.7. Animals and Treatment

#### 3.7.1. Animals

Male Kunming mice weighing 18.0–22.0 g were purchased from Liaoning Changsheng Biotechnology Co., Ltd. (Shenyang, China) and randomly divided into nine groups with 10 mice per group. The groups included control, acute, and subacute model groups and three treatment (25, 50, and 100 mg BRAs/kg BW, suspended with 0.5% carboxymethylcellulose CMC) groups for the two model groups. The animals were maintained under controlled conditions at 22 °C in a specific pathogen-free animal room with a 12 h light/dark cycle and 60 ± 5% humidity. They were allowed free access to tap water and rodent chow. All animal-use procedures were in accordance with the regulations for animal experimentation issued by the State Committee of Science and Technology of the People’s Republic of China and the National Institutes of Health guide for the care and use of laboratory animals (animal ethics number: SCXK (Liao) 2015-0001).

#### 3.7.2. Acute and Subacute ALD Mouse Model Design

The mice were allowed to acclimatize to the environment for 1 week prior to the experiments. The alcohol-induced hepatic injury design followed the method reported by Sun [1]. The experiment lasted for 1 month, and the control group received no intervention. In the acute experiment, 25, 50, and 100 mg BRA/kg BW were intragastrically administered once a day to the low-, middle-, and high-dose groups, respectively, and equal volumes of saline were administered to the model group. The model and treatment groups were treated with 50% ethanol solution (12 mL/kg BW) 4 h of treatment. Then, all groups were fasted for 16 h, and 60 mg/kg BW pentobarbital sodium was injected into the abdominal cavity of each mouse for anesthesia. The food and water intakes of mice were recorded once a day during the experiment.

In the subacute experiment, the model group was administered with saline once daily for 15 days and 30% ethanol (10 mL/kg BW) alone for another 15 days. The low-, middle-, and high-dose groups were treated with BRAs for 30 days, and then 30% ethanol solution was administered once a day for last 15 days to establish the subacute ALD model. The duration between the two administrations was 4 h. Then, all groups were fasted for 4 h, and all mice were anesthetized.

#### 3.7.3. Measurement of Biochemical Parameters

Liver indexes were calculated as a percentage of liver weight/body weight. Blood was sampled from the abdominal aorta to measure the changes in the levels of five serum biomarkers. Serum was obtained through centrifugation after coagulation. AST, ALT, CHO, LDL, and TBIL were tested by using standard clinical laboratory methods and a clinical chemistry analyzer.

#### 3.7.4. Histopathological Observation

All mouse liver sections were fixed in 10% formalin buffer for 48 h. The fixed tissues were embedded in paraffin, cut into 5–6 μm-thick sections, placed on microscope slides for hematoxylin and eosin staining (H & E), and then subjected to microscopic observations (BX51, Olympus, Tokyo, Japan).

#### 3.7.5. Western Blot Analysis

The concentrations of total proteins extracted from the livers after homogenization, centrifugation, and RIPA lysis buffer treatment to rupture cells and proteins were measured by the BCA assay. Liver proteins from each group were separated by 10% SDS-PAGE and then transferred to polyvinylidene fluoride membranes. The membranes were incubated with rabbit antimouse NF-κB and TGF-β antibodies (1:1000) at 4 °C overnight. Blots were treated with the appropriate secondary antibodies at room temperature for 1 h. Positive bands were visualized on X-ray film using ECL reagents (Thermo Fisher, Waltham, MA, USA).

### 3.8. Potential Mechanisms of BRAs against HepG2 Cells

#### 3.8.1. Cell Viability Assay

Cell viability inhibition was examined through the MTT assay. t-HSC/Cl-6, HepG2, Hep3B, and GSE-1 cells were plated in 96-well plates (5000 cells/well). The control cells were exposed to 0.1% DMSO. The test cells were treated with different concentrations of BRAs (25, 50, 100, 150, and 200 μg/mL, dissolved in 0.1% DMSO), cyanidin-*3*-*O*-rutinoside (10, 20, 40, 80, and 100 μM, dissolved in 0.1% DMSO), and positive control (Silymarin and Mitomycin C) for 48 h. After the removal of all residual liquid in the wells, the cells were subsequently incubated with 100 μL fresh medium and 10 µL MTT (5 mg/mL) for 4 h. The reduction in cell viability was determined at 490 nm by using a microplate reader (Bio-Rad iMARK, Berkeley, CA, USA). Three replicates were obtained for each treatment.

#### 3.8.2. Observation of Morphological Changes

HepG2 cells were seeded in a 12-well plate for 24 h of incubation. Then, the cells were treated with BRAs and cyanidin-*3*-*O*-rutinoside for 18 h. The control cells were exposed to 0.1% DMSO.

The cells were fixed with 4% formaldehyde for 30 min at 4 °C. After fixation, the cells were treated with 5 mM DAPI and propidium iodide at 37 °C in the dark for 10 min. Cellular morphology was observed by using a phase contrast microscope (Leica, Nussloch, Germany).

#### 3.8.3. Annexin V-FITC Apoptosis Detection by Flow Cytometry

An Annexin V-FITC apoptosis detection kit was utilized in accordance with the manufacturer’s instructions (BD Biosciences, San Jose, CA, USA) to determine the apoptotic effect induced by BRAs and cyanidin-*3*-*O*-rutinoside. In brief, HepG2 cells were plated in six-well plates (2 × 10^5^/well) and incubated for 12 h. The cells were harvested and washed with PBS through centrifugation after 18 h of treatment with different concentrations of BRAs and cyanidin-*3*-*O*-rutinoside. Then, the cells were resuspended in 100 μL binding buffer and stained with Annexin V-FITC and propidium iodide at room temperature in the dark for 10 min. Thereafter, the cells were assessed by flow cytometry within 1 h after staining, and the data were analyzed with Cell Quest Pro software (BD Biosciences, Franklin Lakes, NJ, USA).

#### 3.8.4. Western Blot Analysis

Cells were harvested after treatment, and RIPA lysis buffer was used to extract the total cellular proteins. Protein concentration was determined through the BCA assay. Equal amounts of protein were separated by 10% SDS-PAGE, transferred to polyvinylidene fluoride membranes, and probed with specific antibodies. Blots were treated with the appropriate secondary antibodies at room temperature for 1 h. Positive bands were visualized on an X-ray film using ECL reagents (Thermo Fisher, Waltham, MA, USA).

### 3.9. Statistical Analysis

All the samples were replicated in triplicate. Two-way analysis of variance (ANOVA) was performed to test the differences between groups, and the mean separation was done by LSD (*p* ≤ 0.05) using SPSS 16.0 program for windows (SPSS Inc., Chicago, IL, USA).

## 4. Conclusions

Alcohol treatment induced a drastic increase in oxidative stress markers, nutritional disturbances, the inflammatory cascade, and histological changes in the liver tissue. Alcohol consumption is an addictive disorder that causes hepatitis, fibrosis, cirrhosis, and cancer. Anthocyanins, a category of natural polyphenols, play an imperative role in health supplements owing to their abundance in fruits and their potential beneficial pharmacological effects.

In this study, BRAs and BRA-derived anthocyanins (purified through column chromatography) exhibited good antioxidant activity as proven by the DPPH, ABTS, and FRAP assays. Four cyanidin anthocyanins, two pelargonidin anthocyanins, and one peonidin anthocyanin were identified through the FT-ICR MS analysis. These results verified the bioactive constituents of BRAs. Network analysis and docking study proved that BRAs have a preventive effect on ALD by inducing apoptosis through the PI3K-Akt signal pathway. The co-administration of BRAs with alcohol considerably suppressed the increased levels of the biochemical parameters ALT, AST, CHO, LDL, and TBIL in vivo. These results showed the protective effect of BRAs against ALD and were affirmed by histopathological observation and liver indexes. This study is the first to demonstrate the protective effect of BRAs against alcoholic liver injury. The mechanisms of this protective effect may be ascribed to the strong antioxidant activities of anthocyanins. In addition, MTT assays showed that anthocyanins exhibited cytotoxic effects on t-HSC/Cl-6, HepG2, and Hep3B. At the same time, BRAs and cyanidin-3-*O*-rutinoside induced the apoptosis of HepG2 cells; downregulated the protein expression level of Bcl-2; upregulated the level of Bax; and promoted the release of cytochrome C, cleaved caspase-9, cleaved caspase-3, and cleaved PARP in HepG2 cells. These investigations provide a new promising natural product as a chemotherapy agent against human hepatocarcinoma and alcohol-induced pathological liver changes.

## Figures and Tables

**Figure 1 molecules-26-02313-f001:**
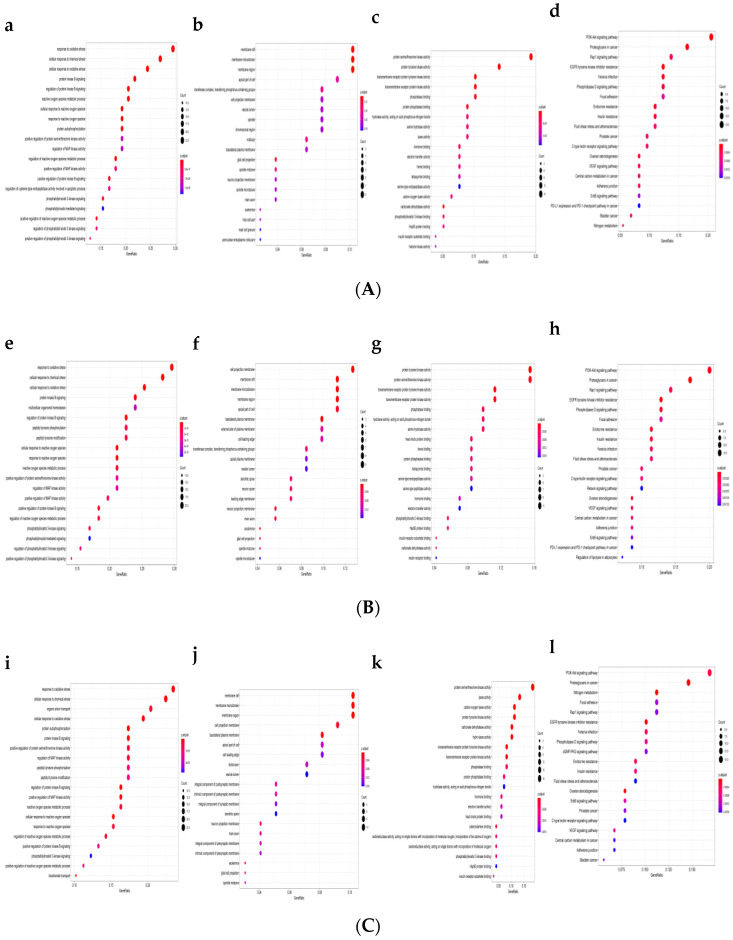
Gene Ontology (GO) and Kyoto Encyclopedia of Genes and Genomes (KEGG) enrichment analysis ((**A**) hepatitis, (**B**) hepatic fibrosis, and (**C**) liver cancer). (**a**,**e**,**i**) The enriched terms in biological process (BP); (**b**,**f**,**j**) the enriched terms in cellular component (CC); (**c**,**g**,**k**) the enriched terms in molecular function (MF); (**d**,**h**,**l**) the enriched terms in KEGG.

**Figure 2 molecules-26-02313-f002:**
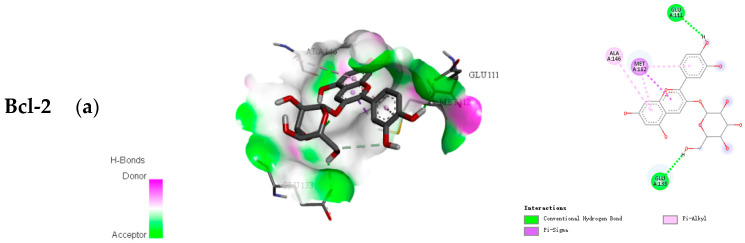
Bioactive compound–protein docking combination: (**a**) Cyanidin-*3*-*O*-glucoside, (**b**) Cyanidin-*3*-*O*-rutinoside, and (**c**) Cyanidin-*3*-*O*-rutinoside).

**Figure 3 molecules-26-02313-f003:**
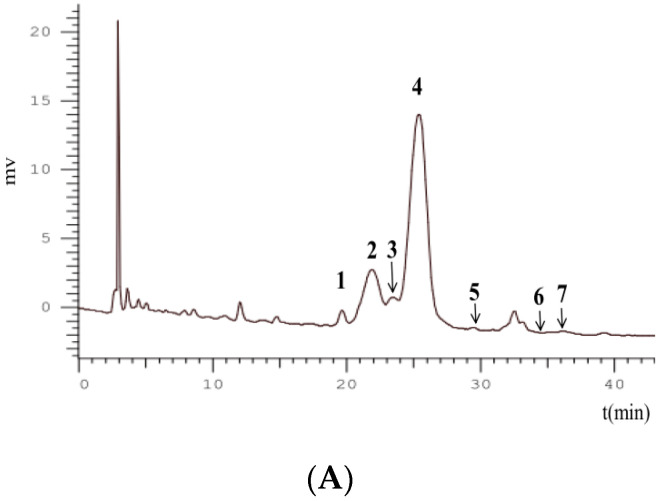
(**A**) HPLC chromatogram and (**B**) structures of black raspberry anthocyanins detected at 280 nm.

**Figure 4 molecules-26-02313-f004:**
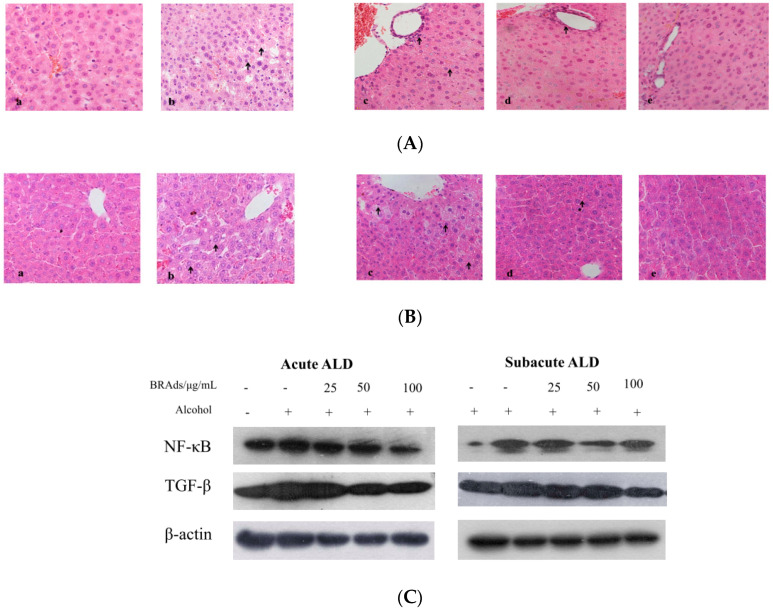
(**A**,**B**) The histological analysis of BRAs on subacute (**A**) and acute (**B**) alcohol-induced liver injury (×500 magnification): (**a**) control group, (**b**) model group, (**c**) low-dose BRA group, (**d**) middle-dose BRA group, and (**e**) high-dose BRA group. (**C**) NF-κB and TGF-β expression in the liver tissue. (**D**) Morphological changes in HepG2 cells. (**E**) The induction of apoptosis was determined by flow cytometric analysis with Annexin V-FITC and PI-staining. (**F**) Effects of cyanidin-*3*-*O*-rutinoside and BRAs on the expression of various cell apoptosis proteins in HepG2 cells.

**Table 1 molecules-26-02313-t001:** Molecular docking mode for anthocyanins with Bcl-2, Caspase 9, and Cytochrome c.

Targets	Compound	Binding Free Energy (kcal/mol)	Numbers of Bonds
Bcl-2	Cyanidin-*3*-*O*-glucoside	−7.3	6
	Cyanidin-*3*-*O*-rutinoside	−8.2	11
	Cyanidin-*3*-*O*-xylosylrutinoside	−8.3	11
Caspase 9	Cyanidin-*3*-*O*-glucoside	−7.6	4
	Cyanidin-*3*-*O*-rutinoside	−8.0	6
	Cyanidin-*3*-*O*-xylosylrutinoside	−8.3	9
Cytochrome c	Cyanidin-*3*-*O*-glucoside	−7.2	5
	Cyanidin-*3*-*O*-rutinoside	−7.7	9
	Cyanidin-*3*-*O*-xylosylrutinoside	−7.7	10

**Table 2 molecules-26-02313-t002:** Identification of phenolic compounds in black raspberry by HPLC-FT-ICR MS/MS and peaks numbers as in Figure 1.

No.	RT (min)	Observed Mass (Da)	Calculated Mass (Da)	Error (ppm)	MS/MS	Formula	Identification	References
1	19.85	581.15071	581.15010	−1.03	287.05478	C_26_H_29_O_15_	Cyanidin-*3*-*O*-sambubioside	[25]
2	21.99	449.10530	449.10784	−0.71	287.05440	C_21_H_21_O_11_	Cyanidin-*3*-*O*-glucoside	[26]
3	23.51	727.20315	727.20811	−1.04	287.05455	C_32_H_39_O_19_	Cyanidin-*3*-*O*-xylosyl-rutinoside	[26]
4	25.78	595.16217	595.16575	−0.23	287.05520	C_27_H_31_O_15_	Cyanidin-*3*-*O*-rutinoside	[26]
5	29.50	433.11031	433.11292	−0.16	271.05945	C_21_H_21_O_10_	Pelargonidin-*3*-*O*-glucoside	[26]
6	34.50	579.16687	579.17083	−1.46	271.05954	C_27_H_31_O_14_	Pelargonidin-*3*-*O*-rutinoside	[26]
7	36.10	609.18196	609.18140	−0.93	301.07027	C_28_H_33_O_15_	Peonidin-*3*-*O*-rutinoside	[27]

**Table 3 molecules-26-02313-t003:** Effect of BRAs on the activity of biochemical parameters.

Treatment Group	Final Weight (g)	Liver Index (%)	AST (U/L)	ALT (U/L)	CHOL (U/L)	TBIL (U/L)	LDL (U/L)
(a) Subacute ALD mice							
Control	35.11 ± 0.91	4.10 ± 0.21	81.40 ± 1.45	28.60 ± 1.56	1.92 ± 0.78	1.23 ± 0.52	0.28 ± 0.18
Model	34.67 ± 0.62 *	5.07 ± 0.47 ***	114.14 ± 2.17 ***	57.71 ± 1.14 ***	2.59 ± 1.01 ***	1.73 ± 1.08 ***	0.48 ± 0.15 ***
Low-dose BRAs	35.18 ± 0.72	4.86 ± 0.22 ^###^	107.13 ± 2.02 ^#^	41.50 ± 1.11 ^##^	2.44 ± 1.13 ^#^	1.45 ± 1.02 ^###^	0.42 ± 0.13
Middle-dose BRAs	35.24 ± 0.64	4.54 ± 0.38 ^###^	101.67 ± 1.67^##^	37.33 ± 1.03 ^###^	2.39 ± 0.76 ^###^	1.31 ± 0.94 ^###^	0.40 ± 0.12
High-dose BRAs	35.12 ± 0.56	4.33 ± 0.32 ^###^	87.13 ± 1.32 ^###^	31.38 ± 0.97 ^###^	2.14 ± 0.81 ^###^	1.26 ± 0.75 ^###^	0.38 ± 0.09 ^#^
(b) Acute ALD mice							
Control	35.14 ± 0.87	4.19 ± 0.22	81.43 ± 1.47	28.58 ± 1.26	1.94 ± 0.88	1.25 ± 0.49	0.29 ± 0.19
Model	34.78 ± 0.57 *	5.26 ± 0.41 ***	133.27 ± 2.74 ***	62.56 ± 1.45 ***	3.31 ± 1.45 ***	1.94 ± 0.98 ***	0.80 ± 0.54 ***
Low-dose BRAs	35.13 ± 0.63	4.95 ± 0.34 ^###^	122.16 ± 2.14 ^##^	52.64 ± 1.32 ^##^	3.06 ± 1.23 ^##^	1.85 ± 0.79	0.73 ± 0.46
Middle-dose BRAs	35.11 ± 0.45	4.73 ± 0.31 ^###^	113.77 ± 1.78 ^###^	44.43 ± 1.13 ^###^	3.01 ± 1.17 ^###^	1.61 ± 0.54 ^#^	0.65 ± 0.43
High-dose BRAs	35.19 ± 0.55	4.29 ± 0.28 ^###^	92.75 ± 1.23 ^###,^*	32.80 ± 1.07 ^###^	2.28 ± 1.02 ^###^	1.33 ± 0.51 ^###^	0.48 ± 0.32 ^#^

Values are the mean ± standard deviation of 10 determinations. Compared with control: *, *p* < 0.05; ***, *p* < 0.001. Compared with model: ^#^, *p* < 0.05; ^##^, *p* < 0.01; ^###^, *p* < 0.001.

**Table 4 molecules-26-02313-t004:** Cytotoxic activity of anthocyanins on HSC, HepG2, Hep3B, and GES-1.

Cell Line	IC_50_ (*n* = 3)			
	BRAs	Cyanidin-*3*-*O*-Rutinoside	Silymarin	Mitomycin C
t-HSC/Cl-6	202.91 ± 10.17 ^a^	17.87 ± 2.43 ^a^ 30.04 ± 3.54 ^b^	192.19 ± 14.22 ^a^	-
HepG2	198.63 ± 9.68 ^a^	13.46 ± 1.78 ^a^ 22.62 ± 2.89 ^b^	-	8.06 ± 1.12 ^a^ 24.10 ± 2.37 ^b^
Hep3B	181.00 ± 12.34 ^a^	9.95 ± 1.81 ^a^ 16.73 ± 2.37 ^b^	-	8.01 ± 1.15 ^a^ 23.96 ± 2.14 ^b^
GSE-1	335.10 ± 18.65 ^a^	37.14 ± 4.07 ^a^ 62.42 ± 9.68 ^b^	-	9.11 ± 1.26 ^a^ 27.26 ± 2.72 ^b^

^a^ μg/mL, ^b^ μM.

## Data Availability

Not applicable.

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
