# Peer review of "Multiple Roles of Black Raspberry Anthocyanins Protecting against Alcoholic Liver Disease"

_molecules, 2021, doi:10.3390/molecules26082313_

Round 1

Reviewer 1 Report

The manuscript by Xia et al. deals with the protecting effect of Black Raspberry anthocyanins on alcoholic liver disease. I consider that the manuscript is interesting, well written and clearly presented. 

I have some minor comments:

- Authors said that: “the BRAs was purified by chromatography, and the content of anthocyanins was higher than ethanol/H2O extracts”.

According to the material and methods section, BRAs is the purified extract obatined by purification of crude EtOH/aqueous extract. Please clarify

Please inform the anthocyanins content in the crude and purified extracts, as well as the detailed  methodology employed to calculate it.

- Authors said that: Two main anthocyanins still were cyanidin-3-O-rutinoside and cyanidin-3-O-gluco-side, and the content were measured by HPLC with 23.71±6.33 mg/g and 3.92±0.71 mg/g, respectively.

To which extract quantification of the anthocyanins refers to?  Is it BRAs?

- Have authors evaluted the effect of the BRAs extract and anthocyanins in normal cells? Have authors determined any selectivity index?

Author Response

Dear editors and reviewers

Thank you very much for the useful suggestions regarding our manuscript entitled “Multiple roles of black raspberry anthocyanins protecting against alcoholic liver disease” (molecules-1171816). The editor's and reviewer's comments are very helpful for revising and improving our paper. We have read all the positive and constructive comments carefully and have tried our best effort to revise the manuscript according to the comments. We hope that the revised manuscript can be considered for publication in molecules.

Revised portion are marked in red in the paper. The main corrections in the paper and the responds to the editor's and reviewer's comments are as flowing:

Point 1: Authors said that: “the BRAs was purified by chromatography, and the content of anthocyanins was higher than ethanol/H2O extracts”.

According to the material and methods section, BRAs is the purified extract obatined by purification of crude EtOH/aqueous extract. Please clarify

Please inform the anthocyanins content in the crude and purified extracts, as well as the detailed methodology employed to calculate it.

Response 1: As the reviewer suggested, we have clarified the anthocyanins content in the crude and purified extracts, as well as the detailed methodology. The added information in this manuscript was marked in red in line 168-171 and line 369-370. (in red)

Point 2: Authors said that: Two main anthocyanins still were cyanidin-3-O-rutinoside and cyanidin-3-O-gluco-side, and the content were measured by HPLC with 23.71±6.33 mg/g and 3.92±0.71 mg/g, respectively.

To which extract quantification of the anthocyanins refers to? Is it BRAs?

Have authors evaluted the effect of the BRAs extract and anthocyanins in normal cells? Have authors determined any selectivity index?

Response 2: Thanks for the reviewer's comments. The quantification of two main anthocyanins refers to BRAs. The added information in this manuscript was marked in red in line 186. (in red)

   We have evaluted the effect of the BRAs extract and anthocyanins in GSE-1 cells, which are normal cells. The cytotoxicity of BRAs, cyanidin-3-O-rutinoside were determined in GSE-1 culture model for 48h of incubation. In the concentrations tested, no significant toxicity was noted. This manuscript was described in line 254-256 and line 443, and the results were shown in Table 4.

Reviewer 2 Report

This reviewer thanks the authors for a comprehensive text that helped to understand the results even for a non-expert on biochemistry and liver diseases. This manuscript presents very valuable research. However, some questions surged after thoroughly reading of the text.

Regarding the section 2.5., the results seem to be in table 3 rather than table 2, as indicated. The authors comment that the liver weight and index were reduced by 13.64% and 14.59%, respectively when high dose BRAs were administered. However, when comparing the values for high-dose and model group, there is a difference between the means of 0.45 g and 0.41 g of increase for liver weight, which represents 1.30% and 1.17%, respectively. Also the differences are lower than the SD (of around 0.55 g). Next, the authors comment that the same results were obtained for both ALDs but, the index decreases a relative 18.44% in acute ALD instead of a 14.59% (subacute), which is better. Also, in this case the differences are higher than the SD of the model and the treatment, which is remarkable. In addition, there is a very small difference for liver weight among the low-, middle- and high-dose BRAs in both ALDs. Therefore, this reviewer recommends that the authors revise this part to comment the results accordingly to what was obtained and written in the table.

Regarding the docking studies, how many iterations were carried out to determine the most favorable conformation with the smallest binding energy? What was the size of the set grid in the binding site used for the docking in Autodock? Were the anthocyanins 3D structure files (pdb?) downloaded from a specific database or were they drawn in ChemDraw and then optimized with an optimization method (semi-empyrical, PM3, ab initio, etc)?

Please, either complete the sections for Supplementary Materials and Author Contributions or remove them if necessary (also for any other empty sections).

Minor corrections: organ (line 40); pigments (line 52); Table 2 and Figure 3 (lines 143–144); peaks (lines 144 and 145); contents (line 152); table 2 last column ‘Reference’; Table 3 (line 160); control (line 229); inhibiting (line 264); Sigma (line 271); MTT name, the H goes in italics (line 276)

Author Response

Dear editors and reviewers

Thank you very much for the useful suggestions regarding our manuscript entitled “Multiple roles of black raspberry anthocyanins protecting against alcoholic liver disease” (molecules-1171816). The editor's and reviewer's comments are very helpful for revising and improving our paper. We have read all the positive and constructive comments carefully and have tried our best effort to revise the manuscript according to the comments. We hope that the revised manuscript can be considered for publication in molecules.

Revised portion are marked in red in the paper. The main corrections in the paper and the responds to the editor's and reviewer's comments are as flowing:

Reviewer 1

Point 1: Authors said that: “the BRAs was purified by chromatography, and the content of anthocyanins was higher than ethanol/H2O extracts”.

According to the material and methods section, BRAs is the purified extract obatined by purification of crude EtOH/aqueous extract. Please clarify

Please inform the anthocyanins content in the crude and purified extracts, as well as the detailed methodology employed to calculate it.

Response 1: As the reviewer suggested, we have clarified the anthocyanins content in the crude and purified extracts, as well as the detailed methodology. The added information in this manuscript was marked in red in line 168-171 and line 371-372. (in red)

Point 2: Authors said that: Two main anthocyanins still were cyanidin-3-O-rutinoside and cyanidin-3-O-gluco-side, and the content were measured by HPLC with 23.71±6.33 mg/g and 3.92±0.71 mg/g, respectively.

To which extract quantification of the anthocyanins refers to? Is it BRAs?

Have authors evaluted the effect of the BRAs extract and anthocyanins in normal cells? Have authors determined any selectivity index?

Response 2: Thanks for the reviewer's comments. The quantification of two main anthocyanins refers to BRAs. The added information in this manuscript was marked in red in line 186. (in red)

   We have evaluted the effect of the BRAs extract and anthocyanins in GSE-1 cells, which are normal cells. The cytotoxicity of BRAs, cyanidin-3-O-rutinoside were determined in GSE-1 culture model for 48h of incubation. In the concentrations tested, no significant toxicity was noted. This manuscript was described in line 256-258 and line 445, and the results were shown in Table 4.

Reviewer 2

Point 1: Regarding the section 2.5., the results seem to be in table 3 rather than table 2, as indicated. The authors comment that the liver weight and index were reduced by 13.64% and 14.59%, respectively when high dose BRAs were administered. However, when comparing the values for high-dose and model group, there is a difference between the means of 0.45 g and 0.41 g of increase for liver weight, which represents 1.30% and 1.17%, respectively. Also the differences are lower than the SD (of around 0.55 g). Next, the authors comment that the same results were obtained for both ALDs but, the index decreases a relative 18.44% in acute ALD instead of a 14.59% (subacute), which is better. Also, in this case the differences are higher than the SD of the model and the treatment, which is remarkable. In addition, there is a very small difference for liver weight among the low-, middle- and high-dose BRAs in both ALDs. Therefore, this reviewer recommends that the authors revise this part to comment the results accordingly to what was obtained and written in the table.

Response 1: We are sorry for our careless about the number of table, we have revised the number of Table. The information in this manuscript was marked in red in line 195. (in red) As the reviewer suggested, we have revised this part to comment the results. The added information in this manuscript was marked in red in line 198-202. (in red)

Point 2: Regarding the docking studies, how many iterations were carried out to determine the most favorable conformation with the smallest binding energy? What was the size of the set grid in the binding site used for the docking in Autodock? Were the anthocyanins 3D structure files (pdb?) downloaded from a specific database or were they drawn in ChemDraw and then optimized with an optimization method (semi-empyrical, PM3, ab initio, etc)?

Response 2: Thanks for the reviewer's comments. In this manuscript, three iterations were carried out to determine the most favorable conformation with the smallest binding energy. The docking in Autodock, X, Y, and Z were set to 6.333, -2.583, and -7.333 Å for Bcl-2; -16.194, -2.667, and 0.444 Å for Caspase 9; -4.028, 23.194, and 38.222 Å for Cytochrome c. Furthermore, the anthocyanins 3D structures were drawn in ChemDraw and then optimized with MM2 method.

Point 3: Please, either complete the sections for Supplementary Materials and Author Contributions or remove them if necessary (also for any other empty sections).

Response 3: Thanks for the reviewer's comments. In this manuscript, Author Contributions are as flowing:

Data curation, Ting Xiao, Zhonghua Luo and Zhenghong Guo; Formal analysis, Ting Xiao; Methodology, Ting Xiao; Project administration, Ting Xiao; Resources, Ting Xiao, Xude Wang, Meng Ding and Wei Wang; Supervision, Xiangchun Shen and Yuqing Zhao; Writing-original draft, Ting Xiao; Writing-review & editing, Ting Xiao.

Point 4: Minor corrections: organ (line 40); pigments (line 52); Table 2 and Figure 3 (lines 143–144); peaks (lines 144 and 145); contents (line 152); table 2 last column ‘Reference’; Table 3 (line 160); control (line 229); inhibiting (line 264); Sigma (line 271); MTT name, the H goes in italics (line 276).

Response 4: We are sorry for our careless of writing errors. We have made corrections about the errors. The information in this manuscript was marked in red in line 64, 77, 178, 179, 185, 187, 223, 225, 227, 228, 233, 244, 272, 274, 286, 307, 308, 316, 321-322, 327, 390 and 794.
